# Low Radiation Doses to Gross Tumor Volume in Metabolism Guided Lattice Irradiation Based on Lattice-01 Study: Dosimetric Evaluation and Potential Clinical Research Implication

**DOI:** 10.3390/jpm15100470

**Published:** 2025-10-02

**Authors:** Giuseppe Iatì, Giacomo Ferrantelli, Stefano Pergolizzi, Gianluca Ferini, Valeria Venuti, Federico Chillari, Miriam Sciacca, Valentina Zagardo, Carmelo Siragusa, Anna Santacaterina, Anna Brogna, Silvana Parisi

**Affiliations:** 1Department of Clinical and Experimental Medicine, Radiation Oncology, University of Messina, 98122 Messina, Italy; giuseppe.iati@unime.it; 2Department of Biomedical, Dental Science and Morphological and Functional Images, University of Messina, 98122 Messina, Italy; giacomo.ferrantelli@outlook.com (G.F.); silvana.parisi@unime.it (S.P.); 3Radiation Oncology Unit, REM Radioterapia, Viagrande, 95029 Catania, Italy; gianluca.ferini@unikore.it (G.F.); valentina.zagardo@grupposamed.com (V.Z.); 4Department of Medicine and Surgery, University Kore of Enna, 94100 Enna, Italy; 5Radiation Oncology Unit, Villa Santa Teresa Hospital, 90011 Bagheria, Italy; valeria.venuti@studenti.unime.it (V.V.); chillari.federico92@gmail.com (F.C.); 6U.O.S.D. Radioterapia Oncologica, Ospedale “Abele Ajello”, 91026 Mazara del Vallo, Italy; miriamsciacca06@gmail.com; 7Medical Physics Unit, A.O.U. “G. Martino”, 98122 Messina, Italy; carmelo.siragusa@polime.it (C.S.); anna.brogna@polime.it (A.B.); 8Radiation Oncology Unit, Papardo Teaching Hospital, 98158 Messina, Italy; anna.santacaterina@virgilio.it

**Keywords:** spatially fractionated radiotherapy, immunotherapy, metabolism-guided radiotherapy

## Abstract

**Purpose**: This paper aims to calculate the gross tumor volume (GTV) receiving low radiation doses in patients submitted to “metabolism-guided” lattice radiation therapy and relative possible implications with clinical outcomes. **Material and Methods**: We reviewed plans for treating voluminous masses via “metabolism-guided” LATTICE-01 irradiation. The aim was to deliver high-dose radiation to spherical deposits (vertices) within a bulky tumor mass. These were placed at tumor areas with differing PET metabolism. We evaluated the relationships between GTV volumes and dose-volumetric histograms (mean, maximum, minimum, and % GTV received 0.5, 1, 2, 3 Gy). **Results**: Sixty-two plans of treatment met the inclusion criteria as established. The median GTV volume was 315.9 cc (range = 10.54–2605.9 cc). A median of two Vertices was allocated within the GTVs (range 1–9) and were planned to receive a dose of ≥10 Gy/1 fraction (median 12 Gy, range 10–15 Gy). Median V3Gy percentage was 51.58% (range 2–100%), median V2Gy percentage was 67.80% (range 1.60–100%), median V1Gy percentage was 83.70% (range 0.80–100%), and median V0.5Gy percentage was 88.49% (range 17.60–100%). **Conclusions**: In the present series, we performed a dosimetric evaluation of the GTV’s volume exposed to low doses during the metabolic guided lattice irradiation process. Combining high- and low-dose radiotherapy based on a spatially fractionated (LATTICE) approach could reactivate the immune system against cancer cells. These observations could be useful for planning prospective studies on immunotherapy combined with the lattice technique.

## 1. Introduction

Advances in radiation therapy technologies permit the delivery of high doses in small volumes, which allows for the “ablation” of both primitive and metastatic tumor deposits. Ablative stereotactic body radiation therapy (SBRT) maintains the same fractional irradiation modality of delivering a homogeneous dose to the entire gross tumor volume (GTV). However, a key difference is the ability to provide less irradiation of surrounding tissues with SBRT. Using spatially fractionated radiation therapy (SFRT), it is possible to treat small volumes within the GTV, providing lower irradiation both to the GTV and surrounding tissues, including “regional” lymph nodes. This technique has been demonstrated to improve clinical outcomes in patients with large non-small-cell lung cancers [1]. Lattice technique, an evolution of GRID one, is a particular form of SFRT that delivers inhomogeneous radiation doses within GTV (“peak dose” and “valley dose”). Using the “classic” lattice technique, usually, vertices (Vs) of dose are allocated geometrically within tumor masses (bulky disease) [2]. We recently reported clinical outcomes in a non-geometrically arranged lattice technique, which uses delivering high doses in small vs. allocated arbitrarily within >5 cm bulky disease on the boundary with areas presenting different fluorodeoxyglucose 18FDG uptake in PET/CT [3], assuming that the 18F-FDG uptake capacity of hypoxic cancer cells may be superior when compared with their normoxic counterparts [4].

These tumor areas are characterized by both more radioresistant tumor cell clusters and a tumor microenvironment (TME) at low immunogenicity that may be prevalent within the tumor mass; this finding has been reported in vivo [5]. With this kind of technique, it is possible delivering both high dose levels, delivered within GTV’ subsites with different metabolic activity on PET-CT imaging and low radiation doses to the gross of tumor mass, which contains lymphatic cells and tumor microvasculature in an immune-cold status, with the aim to reprogram the tumor microenvironment, inducing the simultaneous mobilization of innate and adaptive immunity [6]. It is yet unclear which low doses are able to induce the aforementioned effect, just as it is not possible to ascertain whether the tumor volume covered by these doses could be significant. Furthermore, the detection of low immunogenicity tumoral areas is a debated issue.

We report a dosimetric analysis of the low doses delivered within the GTV in patients submitted to “metabolism-guided” lattice irradiation. The objective of this study is to provide a comprehensive definition of the percentage value of low-dose levels (ranging from 0.5 to 3 Gy), the extent of volume covered by these levels, and the potential impact on TME immunogenicity, according to the clinical response. These preliminary dosimetric data could be used in further clinical analysis.

## 2. Materials and Methods

### 2.1. Eligibility Criteria

We retrospectively reviewed “metabolism-guided” lattice radiation treatment planning of patients treated in our departments, enrolling patients aged 18 and older who presented with bulky mass disease (exceeding 5 cm) arising from solid tumors, classified as either locally advanced or metastatic. These patients were required to be in good clinical condition, as determined by the Eastern Cooperative Oncology Group Performance Status of 2 or below, and to have a life expectancy exceeding three months. We excluded individuals with a documented history of prior radiation therapy to the primary site of the bulky disease. Patients who had received or were planning to receive chemotherapy or immunotherapy/targeted therapy either before or after LATTICE SFRT were eligible. It was a mandatory precondition that the subjects adhere to a seven-day drug washout period. Pretreatment evaluation was performed with physical examination and complete blood count. All patients were studied with computed tomography (CT) scans of the head, neck, thorax, abdomen, and pelvis. A [18F]-fluorodeoxyglucose positron emission tomography/computed tomography (18F-FDG-PET/CT) scan was conducted. MRI was performed when necessary. We deemed any lesion unsuitable for surgical resection or ablative stereotactic irradiation after a multidisciplinary evaluation.

Informed consent of all patients was acquired, and the study was conducted in accordance with the Declaration of Helsinki and approved by the Messina Ethics Committee of AOU Policlinico “G. Martino” with protocol code 1611-38-21.

### 2.2. Plan Contouring and Treatment Criteria

Treatment was planned using a PET imaging-guided LATTICE method as previously described [3].

CT simulation images with a thickness of 1.25 mm were acquired for all patients immobilized in treatment position, taking into account the anatomic site involved. In head and neck cancers, we employed a thermoplastic mask. In thoracic cancer, patients were positioned with their arms above their heads, using a breast board device. In abdominal pelvic cancer, we used a knee and foot fix for the treatment planning process simulation; the CT was co-registered with the 18F-FDG-PET-CT. The Bulky-Gross Tumor Volume (B-GTV) was delineated. A minimal expansion of GTV (3–5 mm) was performed to obtain the CTV. The PTV definition exhibited variability according to the LinAc equipment, ranging from 2 to 12 mm of isotropic margin expansion, which could be manually adjusted to spare any neighboring critical organs at risk (OARs). In cases of lesions subject to organ motion, we performed tumor motion tracking. Set-up verification was conducted daily, based on the LINAC employed to deliver the treatment. We used an orthogonal anterior–posterior/lateral low-energy X-rays for Cyberkinife treatments and an MV pair and cone beam CT in cases of IMRT/VMAT treatments.

All treatments were planned and delivered in two phases. The first involved the LATTICE technique with the single-shot vertices irradiation. Briefly, a variable number of virtual spheres with a diameter of 10 mm (0.52 cc) were employed and allocated within the mass in an arbitrary, non-geometrical fashion between areas with ‘avid’ 18FDG/PET metabolic activity showing SUV > 2.5 and ‘super avid’ areas with different metabolic activity with SUV 75% compared to SUV max.

We have proposed that different SUVs within the bulky mass could have different rates of cell growth, oxygenation, and tumor microenvironment. The Vs’ position (median 4, range 1–6) took into account the adjacent OARs and the mass volume. The vertices had to have at least a 2.0 cm (center to center) distance from each other.

The aim was to deliver on the vs. a single fraction of ≥10 Gy (range 10–15 Gy) using both stereotactic and non-stereotactic irradiation planned with an isocentric approach in the case of Cyberknife treatment or with an isocentric approach in the case of Intensity Modulated Radiation Therapy (IMRT) and/or Volumetric Modulated Arc Therapy (VMAT). From this analysis, the plans that work out the delivery of more than a fraction to the Vs. The second phase has involved the PTV irradiation, and doses varying from 20 Gy/4 F to 30 Gy/10 F were delivered with a non-stereotactic modality. The choice of fractionation was based on the tumor volume and the tolerance of the proximal OARs.

### 2.3. Treatment Planning Setting

The non-stereotactic treatment plans for all patients were optimized to be delivered using the Versa HD linear accelerator with X-ray beams of 6 and 10 MV in a flattening filter-free mode (FFF). VMAT plans were created using the Monaco Treatment Planning System (TPS, Elekta Sweden, version 6.1.2.0), which allows for different biological cost functions and accurate calculations in complex geometries. For each patient, the best configuration of parameters was determined based on their anatomical site, the number, and the location of the Vs. In single vs. plans, the sphere’s center was used as the isocenter, while for multiple Vs, a target volume was created, and the center of this volume was selected as the isocenter. The treatment planning was performed with dual arcs: a 340° arc for centrally located Vs and a 180° arc for laterally positioned Vs. The collimator angle was generally fixed at 0°, with a 1% statistical uncertainty for calculations. The VMAT optimization used specific parameters to control dose modulation, with an Increment (Inc.) (North Sydney, NSW, Australia) value of 30° found to be optimal for this study.

Stereotactic treatments were delivered using Cyberknife System (CKS, Accuray Inc., Sunnyvale, CA, USA, 6MV FFF), a robotic radiosurgery dedicated device with submillimeter spatial resolution; beams are delivered using fixed circular beams or dynamic variable—aperture Iris collimator. Single or multiple vs. treatment plans were carried out using Precision TPS (Accuray, version 3.3.0.0), employing multiple Iris or fixed collimator and VOLO^TM^ algorithm (inverse planning) that simultaneously optimizes multiple objectives, creating a single multi-criteria cost function, without the availability of absolute constraints. By employing a stereotactic prescription, it was possible to enhance dose heterogeneity within the Vs up to 120–125%.

### 2.4. Dosimetric Evaluation

The radiation dose prescription was performed according to the International Commission on Radiation Units and Measurements (ICRU) recommendations. The planned dose to be delivered to the vertices was at least 10 Gy/1fraction; the optimized plan had to result in ≥98% dose coverage of the vertices volume. Special emphasis was made to avoid overlap of the 8 Gy isodoses of two consecutive vertices, with the objective of preventing the occurrence of excessive hot spots, particularly in the vicinity of OAR.

We performed an analysis of the dose volume histograms in every vertex plan and we registered the percentage of tumor mass volume which received doses of 0.5 Gy, 1 Gy, 2 Gy and 3 Gy (V0.5 Gy, V1 Gy, V2 Gy, and V3 Gy), the minimum dose (Dmin), the maximum dose (Dmax), and the mean dose (Dmean) to the GTV. Finally, an analysis was conducted on the proportion of the GTV volume exclusively exposed to low-dose irradiation, specifically within the intervals 0.5–1 Gy (ΔV_0.5–1Gy_), 1–2 Gy (Δ_V1–2Gy_), and 2–3 Gy (ΔV_2–3Gy_).

### 2.5. Treatment

LATTICE treatment was delivered in a single stereotactic session on day 1, then, after 3 days, we treated the PTV with 4 to 10 consecutive fractions (in this case, the standard fractionation scheme was applied with five sessions per week, with a weekend interruption). This scheme employed non-stereotactic IMRT/VMAT techniques. The theoretical objective is to awaken the immune system in the first phase by remodeling the TME and then complete the treatment by delivering tumor-killing doses to the rest of the target.

### 2.6. Endpoints

In this series, patients with homogeneous characteristics were recruited, and a dosimetric evaluation of low doses within bulky disease was performed. The future aim of our study is to ascertain whether there is a correlation between the target volume covered by a given dose and the clinical response that may be related to immunogenic reactivation.

Here, we show a preliminary report of our analysis. Once statistical significance has been achieved with a greater number of patients, more comprehensive data will be published.

## 3. Results

From October 2018 to September 2024, we evaluated 66 treatment plans of 67 patients, and 62 met the inclusion criteria. We treated patients with metastatic cancer from different histologies, including 11 lung, 11 head and neck, 8 breast, 8 bladder, 5 sarcomas, 5 melanomas, 3 prostate, 3 rectal, 3 kidney, 1 ovarian, 1 mesothelioma, and 2 unknown primary cancers. A total of 36 patients (50%) were being treated with immunotherapy, but in progressive disease, and during and after the radiation treatment, they maintained the same systemic therapy. Regarding the radiotherapy, a median of two Vertices was allocated within the GTVs (range 1–9): 1 vertex was planned in 30 cases; 2 vs. in 13; 3 vs. in 8; 4 vs. in 7; 5 vs. in 3; 9 vs. in 1 case. The median GTV volumes were 315.9 cc (range = 10.54–2605.9 cc). In all cases, we delivered to the vs. doses of ≥10 Gy/1 fraction (median 12 Gy, range 10–15 Gy). With regard the volume of GTV covered by low doses, according the critical aspect of the study, the median percentage of V3 was 51.58% (range 2–100%), the median percentage of V2 was 67.80% (range 1.60–100%), the median percentage of V1 was 83.70% (range 0.80–100%), and the median percentage of V0.5 was 88.49% (range 17.60–100%). Median Dmin to GTV was 0.14 Gy (range 0–1.08 Gy), median Dmax was 18.02 Gy (range 9.51–23.68 Gy), median Dmean was 3.14 Gy (range 0.78–9.33 Gy). The median ΔV_0.5–1Gy_ was 5.9%, the median ΔV_1–2Gy_ was 9.16%, while 11.35% was the median ΔV_2–3Gy_. The total amount of GTV receiving doses only from 0.5 Gy to 3 Gy is 26.41% The treatment was completed with the irradiation of the entire bulky disease, which started 3 days after the vs. irradiation. The median dose delivered to the PTV was 20 Gy/4 fractions (range 20 Gy/4 fr–30 Gy/10 fr).

Figure 1 shows an example of a “metabolism-guided” LATTICE treatment planning in a patient with regionally advanced squamous cell head and neck cancer in progressive disease under immunotherapy with cemiplimab. The dose delivered to the four vertices was 10 Gy. The treatment was completed with a dose of 20 Gy in four fractions (5 Gy/dd) on GTV, delivered 3 days from the first phase. Figure 2 shows the complete disease remission six months after the treatment. The patient is currently continuing the same systemic therapy with no further progressive disease.

## 4. Discussion

Several factors have been identified as being able to influence the activity of the immune system within the tumor mass.

Firstly, it has been demonstrated that tumor cells can create a microenvironment that is inhibitory to immune system function. This may include the production of immunosuppressive molecules and cytokines, such as transforming growth factor beta (TGF-β), which alter immune cell activation and function [7]. Secondly, the ability of tumor cells to reduce the antigen’s presentation to T cells due to an upregulation of the major histocompatibility complex (MHC), a crucial step in the activation of the immune response, has been demonstrated [8].

Furthermore, tumor cells can express proteins that interact with immune checkpoints, such as PD-L1. The binding of PD-L1 to the PD-1 receptor on T cells results in the inhibition of their activation and response against the tumor. This phenomenon can be considered a ‘self-protection’ mechanism of the tumor [9]. Another salient factor to consider is that tumors exhibit irregular vascularization, which prevents the access of immune cells to the tumor core [10]. Finally, the process of acidification of the tumor microenvironment, frequently induced by elevated levels of anaerobic metabolism, has been demonstrated to result in a reduction in the effectiveness of immune responses [11]. The aforementioned factors contribute to the development of a ‘barrier’ that hinders the immune system’s capacity to recognize and destroy tumor cells effectively.

The field of research focused on overcoming the limitations of the immune system in tumors is currently under active investigation, and several therapeutic approaches have been developed to enhance or restore the immune system’s ability to recognize and destroy cancer cells. Immune checkpoint inhibitors, including nivolumab (anti-PD-1), pembrolizumab (anti-PD-1), atezolizumab (anti-PD-L1), and ipilimumab (anti-CTLA-4), are drugs that block these interactions and “unlock” T cells, thereby allowing them to attack the tumor [12,13]. Another therapeutic option is represented by CAR T-cell treatments, in which the patient’s T cells are genetically engineered to express chimeric receptors that recognize specific antigens on the surface of tumor cells [14]. Furthermore, the modification of the tumor microenvironment through the use of inhibitors targeting cytokine signalling pathways, such as TGF-β and antiangiogenesis inhibitors, such as bevacizumab, can render it more susceptible to the action of the immune system [15,16].

Cytokines have been employed as immunological stimulants to augment immune responses, thereby promoting T-cell proliferation and activation against neoplastic cells [17].

The combination of low-dose radiotherapy (LDRT) and intratumoral immunomodulation is an emerging therapeutic theory that aims to harness the power of the immune system to attack cancer cells. The potential of these two modalities to enhance antitumor immune responses has been individually demonstrated, and their combination is being explored for synergistic effects [18].

LDRT is characterized by radiation doses that are lower than those employed in conventional cancer treatment, typically ranging from 0.5 to 2 Gy per fraction, in contrast to standard doses of 2–3 Gy per fraction.

The immunomodulatory effects within the tumoral mass of LDRT are as follows:The immune system is activated, inducing a modest inflammatory response at the tumor site. This process subsequently results in the activation and recruitment of immune cells, including T cells, natural killer (NK) cells, and dendritic cells, and in the decrease of immune-suppressing cells (e.g., regulatory T-cells, myeloid-derived suppressor cells, and TAM M2).Furthermore, LDRT has been demonstrated to stimulate the release of tumor-associated antigens, thereby rendering tumor cells more recognizable to the immune system.LDRT has been demonstrated to promote a form of immunogenic cell death through selective induction of apoptosis or senescence in aberrant cells, thereby helping to “train” the immune system against the cancer cells. The TME encompasses three principal components: cellular elements (fibroblasts, stromal, immune, and endothelial cells), extracellular matrix proteins, and soluble factors. Within this complex niche, both innate immune cells (macrophages, mast cells, neutrophils, dendritic cells, myeloid-derived suppressor cells, and NK cells) and adaptive immune cells (T and B lymphocytes) are present. Their interactions with tumor cells regulate growth, immune evasion, angiogenesis, and metastasis. Based on these assumptions, it is possible to resume some critical points, which are fundamental to understand the immunogenic effect caused (also) by radiation therapy: (A) The presence of tumor antigens (initiation of disease) is retained in memory by T and B lymphocytes; (B) Progression and the formation of metastases are caused by the immune system escaping the tumor; (C) The tumor’s microenvironment and the area ‘around’ GTV contains T lymphocytes (cytotoxic CD8, CD4 Helper, and FOXP3 regulators) activating or inhibiting the immune system [19].

It is hypothesized that this TME homeostasis could be influenced by an oscillation of both high-dose (‘’peak dose’’) regions T and the low-dose (‘’valley dose’’) ones that are homogeneously scattered within the disease, as obtained with SFRT. The effects of LDRT up to 2 Gy per fraction in cancer immunotherapy have not been fully explored. Early evidence from a mouse model of locald neuroendocrine pancreatic tumors suggested that low-dose irradiation (0.5–2 Gy) can reprogram the TME, inducing iNOS-positive M1 macrophages, which produce relevant chemokines to recruit effector T cells. These cells then infiltrate the tumor and trigger a range of mechanisms, which eliminate the tumor cells through the activation of Nk and other native immune cells [6]. Furthermore, the differentiation of effector T cells into memory T cells has been shown to provide long-term surveillance, thereby reducing the risk of tumor recurrence [20]. High-dose radiotherapy, instead, cause ‘damage’ which activates the innate (rapid) response enabling the release of danger-associated molecular patterns that lead to the recruitment of immune cells, including activation of APC cells (e.g., dendritic cells) thus inducing a systemic response against tumor antigens that contrasts local progression of the cancer cells and also mediates distant antineoplastic effects [21]. In contrast, conventional radiation’s doses, homogenously scattered around the target volume induces local immunosuppression by a variety of mechanisms, as: (1) impaired recruitment of immune effector cells through disrupted and/or altered vessels [22,23]; (2) polarization of tumor-associated macrophages towards an immunosuppressive M2-like phenotype [24,25]; conversion of exhausted T cells into immunosuppressive regulatory T (Treg)-like cells [26,27]; (4) activation of CD4+CD25+FOXP3+ Treg cells and immunosuppressive dendritic cell populations [28]; (5) repression of tumor-targeting γδ T cells [29]; and (6) downregulation of MHC molecules on the surface of malignant cells [30]. Consequently, a subsequent (slow) adaptive response may be deficient if lymphocytes surrounding metastases or regional lymph node germinal centres absorb doses greater than 2–3 Gy [31].

SFRT involves an inhomogeneous dose distribution within the tumor, alternating with peaks and valleys in doses. SFRT challenges traditional radiobiological dogma by leveraging the immunomodulatory aspects of radiotherapy to achieve tumor control, exploiting the effect of high-dose ‘peaks’ that ablate microvasculature, immunosuppressive cells, and tumor cells, leading to catastrophic DNA damage, immunogenic cell death, inflammation and the release of neoantigens. Cytokines and TAA secreted from high-dose regions and scatter radiation delivered to low-dose regions can induce bystander [32] cell death and remodel the tumor microenvironment. These changes facilitate the migration of immune cells (innate and adaptive effector cells, including CD8+ cytotoxic T lymphocytes (CTLs), T helper 1 (TH1)-polarized CD4+ T cells and NK cells) [6] to “colder” intratumoral areas, converting them into ‘hot’ ones. It is also noteworthy that the changes in the low-dose ‘valleys’ facilitate the migration of antigen-presenting cells to the border of high-dose regions. In this region, the antigen-presenting cells collect neoantigens from ablated tumor cells and subsequently migrate to nearby tumor-draining lymph nodes. In these lymph nodes, the antigen-presenting cells prime naive T cells to tumor neoantigens [33,34].

Furthermore, this lymph node activation seems to be the basis for inducing the abscopal effect, eliciting robust T and NK cell infiltration at distant non-irradiated tumor sites in patients with oligometastatic solid cancers [35].

The identification of the immunorefractory areas is of crucial importance, and the use of metabolic imaging could assist in the optimization of our treatment. FDG uptake and tumor hypoxia are closely interconnected phenomena that play significant roles in tumor biology and cancer imaging, particularly in PET scans. Tumor hypoxia has been demonstrated to induce a metabolic shift toward glycolysis, even under conditions of oxygen scarcity. FDG uptake frequently correlates with hypoxic regions but is not exclusive to them. Hypoxic regions, in fact, may exhibit low FDG uptake due to severe acidosis or necrosis. Conversely, normoxic tumor areas with high metabolic activity may also demonstrate increased FDG uptake. The role of hypoxia in radiation therapy resistance is well-documented, with oxygen being a prerequisite for effective free radical generation. Consequently, the combination of FDG-PET with hypoxia imaging agents (e.g., ^18F-fluoromisonidazole or ^18F-FAZA) can provide complementary information about metabolic activity and oxygenation [36].

The described case in this paper confirms the clinical and experimental data on the immunogenic activation linked to the “high and low dose” efficacy with a dramatic clinical tumor response.

This is indirect evidence that tumor regression is due to immunological reactivation of the TME after progressive disease during cemiplimab administration.

There are findings to suggest that the reactivation of the antitumor response is an intrinsic factor. This has been identified as the 6R of radiobiology and is strictly influenced by low- and high-dose combinations, which can stimulate immune cells [37]. To date, the effective doses needed to induce this effect are unknown.

Our group has the most extensive case series in the literature of spatially fractionated treatments under “metabolic guidance”, with dose vertices positioned in a non-geometric fashion. This approach seems to deliver a precise and personalized irradiation.

Finally, the objective of our research is to try a low dose percentage level, which should be more effective in stimulating immunogenicity.

### 4.1. Limits

The principal limitation of the study is the lack of robust clinical data. This is a dosimetric study on a patient’s cohort, not sufficient for a correct statistical analysis. To date, we are performing an analysis in the entire group of patients (more than 100) treated with metabolism-guided irradiation that will permit a more accurate statistical analysis.

### 4.2. Future Development and Remarks

The spatial volume exposed to LDRT within the tumor may be critical for shaping the immunogenic outcome. A broader distribution of low-dose regions across the tumor mass could increase the probability of modifying the immunosuppressive milieu within peripheral tumor zones, where immune–tumor interactions are most active. This may facilitate infiltration of effector T cells and dendritic cells, creating a more permissive environment for immune priming. Conversely, a limited or highly localized volume of LDRT might restrict these effects to a small region, reducing the likelihood of systemic immunogenicity. It suggests that treatment planning in radiotherapy should not only consider dose intensity and conformity but also the extent of tumor volume exposed to low doses. Such an approach may optimize the synergy of radiotherapy with immunotherapies such as immune checkpoint inhibitors, potentially enhancing systemic tumor control through the so-called bystander effect. Moreover, regarding the “hypoxic effect”, it is possible to hypothesize that the use of low-dose radiation provides a novel strategy for addressing the challenges posed by hypoxic tumors by reprogramming the tumor microenvironment, normalising vasculature, and enhancing immune infiltration. By mitigating the effects of hypoxia, LDRT may have the potential to enhance tumor sensitivity to radiation, immunotherapy, and other treatments, which may result in improved clinical outcomes.

## 5. Conclusions

In the present series, GTV’s volume exposed to low doses (>0.5 Gy <3 Gy) during the metabolic guided LATTICE irradiation is consistent (>25%). Our early clinical experience suggests that combining high- and low-dose radiotherapy based on a spatially fractionated (LATTICE) approach could reactivate the immune system against cancer cells, especially in patients who are resistant to immunotherapy.

A comprehensive analysis of our data should confirm the clinical impact of low dose within the GTV. The targeting of low-dose radiation to areas of low immunogenicity, identified using metabolic imaging, could be a valuable approach in the planning of prospective trials of immunotherapy combined with altered fractionated radiotherapy as spatially fractionated techniques.

## Figures and Tables

**Figure 1 jpm-15-00470-f001:**
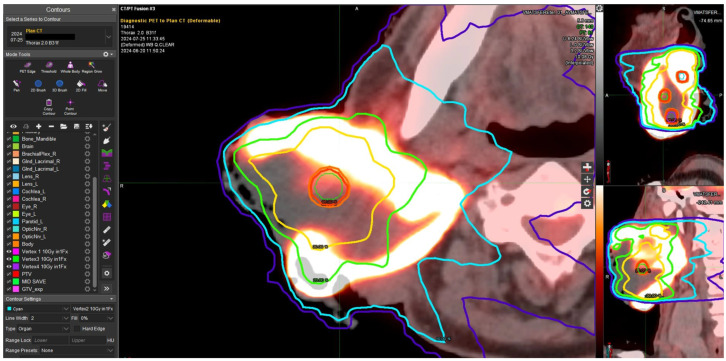
The following example illustrates a LATTICE treatment planning. The isodose curves, which are employed to induce the potential immunomodulatory effect, are illustrated here. Vertices were allocated between areas, with the “Avid” area showing SUV 2.5 and a “super Avid” area with SUV 75% with respect to SUV max. The dark blue line indicates 50 cGy, the light blue line indicates 100 cGy, the green line indicates 200 cGy, and the orange line indicates 300 cGy. The isodose prescription (10 Gy) is represented by a red line.

**Figure 2 jpm-15-00470-f002:**
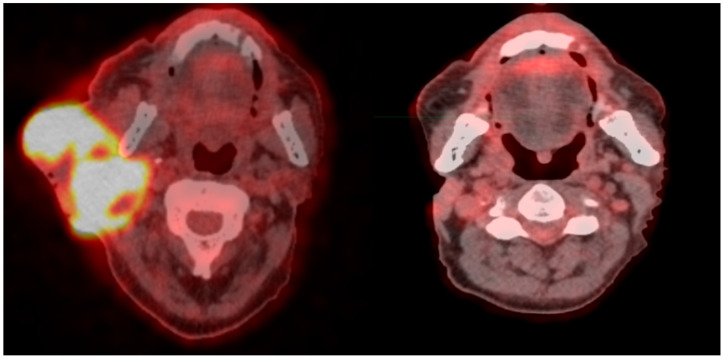
The figure shows the complete response after radiotherapy. The patient is maintaining the therapy with cemiplimab.

## Data Availability

The raw data supporting the conclusions of this article will be made available by the authors on request.

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
