# Peer review of "Low Radiation Doses to Gross Tumor Volume in Metabolism Guided Lattice Irradiation Based on Lattice-01 Study: Dosimetric Evaluation and Potential Clinical Research Implication"

_jpm, 2025, doi:10.3390/jpm15100470_

Round 1
Reviewer 1 Report
Comments and Suggestions for Authors
Thank you to the Authors for the manuscript submitted
The author concludes that a significant portion of the gross tumor volume (GTV) receives low radiation doses during metabolism-guided lattice irradiation. ​ Targeting low-dose radiation to areas of low immunogenicity, identified through imaging techniques, could be valuable for planning prospective studies combining immunotherapy with the lattice technique. ​ These observations could be useful planning prospective studies on immunotherapy combined with lattice technique.
I read with interest the paper (potentially it is of interest) however I found several weak points that need to be clarify
“Briefly, vertices were allocated 73 arbitrarily in a non-geometrical fashion between areas with different 18FDG/PET metabolic activity; “: Considering the paper and potential audience it is important to describe how the Vertices were placed (no grid, vertex diameter , where and how they are placed: based on PET SUVmax, use of cold vertices for planning)
Figure: it is not clear where are the vertices
M&M:
Witch kind of tumor population? Any combination to immunotherapy?
The planned dose to be delivered to the vertices was at least 10Gy/1fraction; add the range as in results
In M&M and results: Analyzing dose volume histograms in every plan, we registered the percentage of tumor mass volume which received doses of 0.5Gy, 1Gy, 2Gy and 3Gy (V0.5, V1, 84 V2 and V3Gy). Median V3 percentage was 51.58% (range 2%-100%), median V2 percentage was 67,80% (range 60% 100%), median V1 percentage was 83,70% (range 0,80%-100%) and median V0.5 percentage was 88,49% (range 17.60%- 100%).
Considering the treatment planning of the single shot of 10Gy to the vertex/vertices?
Please check the range of %: 100% of 1 or 2 or 0.5 Gy to the tumor volume does not sound to me, please check or clarify
Add Gy after V1, 2 3 0,5
The authors present they retrospective eexperience with no clinical outcome on control.
During treatment planning low doses was a clinical goal? It is not clear to me
The discussion is more a narrative review of immune effect of low doses than a discussion in regard to lattice approach
The conclusion are not supported by data (no clinical results only dosimetric)
The Authors should start from the concept of low dose and clarify they try to deliver both high does for control and lo doses for immunogenic effects as aim of the study (if this is the aim). Add clinical results if any.
And in discussion describe the how lattice my help doing this
Author Response
Comment 1:
Thank you to the Authors for the manuscript submitted
The author concludes that a significant portion of the gross tumor volume (GTV) receives low radiation doses during metabolism-guided lattice irradiation. Targeting low-dose radiation to areas of low immunogenicity, identified through imaging techniques, could be valuable for planning prospective studies combining immunotherapy with the lattice technique. These observations  could be useful planning prospective studies on immunotherapy combined with lattice technique.
I read with interest the paper (potentially it is of interest) however I found several weak points that need to be clarify
“Briefly, vertices were allocated 73 arbitrarily in a non-geometrical fashion between areas with different 18FDG/PET metabolic activity; “: Considering the paper and potential audience it is important to describe how the Vertices were placed (no grid, vertex diameter , where and how they are placed: based on PET SUVmax, use of cold vertices for planning).
Figure: it is not clear where are the vertices
 Response:
I thank the reviewer for pointing this out. We agree with this comment. The explanation of the technique is provided in the Materials and Methods section. See Matherials and Method, page 3, lines 87-90. We improved figure and add images pre and post treatment.
M&M:
Comment 2: Witch kind of tumor population? Any combination to immunotherapy?
Response: In page 3, lines 108-112, we specified the tumour population and the type of systemic treatment.
Comment 3: The planned dose to be delivered to the vertices was at  least 10Gy/1fraction; add the range as in results
Response: The range is specified in page 3 line 91
Comment 4: In M&M and results: Analyzing dose volume histograms in every plan, we registered the percentage of tumor mass volume which received doses of 0.5Gy, 1Gy, 2Gy and 3Gy (V0.5, V1, 84 V2 and V3Gy). Median V3 percentage was 51.58% (range 2%-100%), median V2 percentage was 67,80% (range 60% 100%), median V1 percentage was 83,70% (range 0,80%-100%) and median V0.5 percentage was 88,49% (range 17.60%- 100%).
Considering the treatment planning of the single shot of 10Gy to the vertex/vertices?
Please check the range of %: 100% of 1 or 2 or 0.5 Gy to the tumor volume does not sound to me, please check or clarify
Add Gy after V1, 2 3 0,5
 Response: Thank you for your suggestion; we added Gy after V. We considered V0.5-3 as the dose to a percentage of the total GTV volume. Page3 line 101
Comment 5: The authors present they retrospective experience with no clinical outcome on control.
During treatment planning low doses was a clinical goal? It is not clear to me
The discussion is more a narrative review of immune effect of low doses than a discussion in regard to lattice approach
The conclusion are not supported by data (no clinical results only dosimetric)
The Authors should start from the concept of low dose and clarify they try to deliver both high does for control and lo doses for immunogenic effects as aim of the study (if this is the aim). Add clinical results if any.
And in discussion describe the how lattice my help doing this
Response: The purpose of the present study is to ascertain the existence of a potential correlation between the volume affected by a given dose and the clinical response, as indirect evidence of immunological reactivation. The correlation of this reactivation with techniques that allow heterogeneous dose distribution within the tumour, with peaks of high doses and valleys of doses down to 3 Gy as lattice, GRID etc., is well documented in the literature (see ref. 34). The assertion is that, even when taking into account the differential radiosensitivity exhibited by distinct immune cell subtypes (e.g., lymphocytes and macrophages), the potential efficacious doses remain merely conjectural. We added a sentence on this issue in lines 350-353.
This is a dosimetric study on a limited patients’ cohort not sufficient for a correct statistical analysis. Up-to-date we are performing an analysis in the entire group of patients (more than 100) treated with metabolism guide irradiation that will permit a more accurate statistical analysis.
Reviewer 2 Report
Comments and Suggestions for Authors
This is an extremely interesting paper that attempts to explain the biological effects of low-dose radiation therapy and its influence on the functioning of the immune system. Both the theory and treatment algorithm are particularly exciting for both specialists and non-specialist readers. The detailed immuno-oncological and radiation biology discussion is another real special value of the paper.
Despite all the above, the subject of the present research, and the relationship between the methodology, the observed results and the conclusion, is not clear to the reader (to the reviewer). It is evident that during palliative radiation treatment of a bulky tumor, if certain parts of the tumor are subjected to a single high-dose SBRT treatment, with designating certain target volume regions within the tumor, then other parts of the tumor will certainly receive lower-dose (1Gy, 2Gy etc.) irradiation. It is not surprising. Moreover, this finding alone does not necessarily prove the theory, since all the patients received standard radiotherapy as well, after the first, specialized radiation treatment.
Therefore, it would be necessary to include the clinical aspect and clinical interpretation in the discussion (the authors have a clinical database with the details of RT, clinical and radiology follow-up and treatment outcomes and they have previous clinical-based works in this topic). Only in this way would the dosimetry data and the tumor biology discussion become interpretable. Therefore, for my part, I would like to suggest restructuring the manuscript, re-distributing the final message (e.g. special radiobiology, immuno-oncology approach of combined low-dose + high-dose RT with some clinical experiences), and even rewriting the abstract and the conclusion.
Despite all these criticisms and suggestions, the work and the basic idea are extremely valuable and professionally progressive (and it belongs to “personalized medicine”), and the manuscript is recommended to re-evaluate after reconstruction of the work.
Author Response
Comment:
This is an extremely interesting paper that attempts to explain the biological effects of low-dose radiation therapy and its influence on the functioning of the immune system. Both the theory and treatment algorithm are particularly exciting for both specialists and non-specialist readers. The detailed immuno-oncological and radiation biology discussion is another real special value of the paper.
Despite all the above, the subject of the present research, and the relationship between the methodology, the observed results and the conclusion, is not clear to the reader (to the reviewer). It is evident that during palliative radiation treatment of a bulky tumor, if certain parts of the tumor are subjected to a single high-dose SBRT treatment, with designating certain target volume regions within the tumor, then other parts of the tumor will certainly receive lower-dose (1Gy, 2Gy etc.) irradiation. It is not surprising. Moreover, this finding alone does not necessarily prove the theory, since all the patients received standard radiotherapy as well, after the first, specialized radiation treatment.
Therefore, it would be necessary to include the clinical aspect and clinical interpretation in the discussion (the authors have a clinical database with the details of RT, clinical and radiology follow-up and treatment outcomes and they have previous clinical-based works in this topic). Only in this way would the dosimetry data and the tumor biology discussion become interpretable. Therefore, for my part, I would like to suggest restructuring the manuscript, re-distributing the final message (e.g. special radiobiology, immuno-oncology approach of combined low-dose + high-dose RT with some clinical experiences), and even rewriting the abstract and the conclusion.
Despite all these criticisms and suggestions, the work and the basic idea are extremely valuable and professionally progressive (and it belongs to “personalized medicine”), and the manuscript is recommended to re-evaluate after reconstruction of the work.
Response
Thank you for pointing this out
This is a dosimetric study on a limited patients’ cohort not sufficient for a correct statistical analysis. Up-to-date we are performing an analysis in the entire group of patients (more than 100) treated with metabolism guide irradiation that will permit a more accurate statistical analysis.
We have revised the manuscript in all sections and reformuled in part discussion and conclusion as you suggested.
Reviewer 3 Report
Comments and Suggestions for Authors
The major part of the manuscript consists of a Discussion of various aspects of immunotherapy. This is an interesting effort which might be published as a review.
Results are limited to an example of LATTICE treatment planning, without any detail on the type of tumor or the clinical context. The results offer no evidence that the example shown in Figure 1 has any relevance for the immunological phenomena brought up in the Discussion.
Author Response
Comment:
- The major part of the manuscript consists of a Discussion of various aspects of immunotherapy. This is an interesting effort which might be published as a review. 
- Results are limited to an example of LATTICE treatment planning, without any detail on the type of tumor or the clinical context.
- The results offer no evidence that the example shown in Figure 1 has any relevance for the immunological phenomena brought up in the Discussion.  
Response
I thank the reviewer for the observation.
- We tried to ameliorate the entire paper.
- Figure has been ameliorated; besides we add two figures.
- This is a dosimetric study on a limited patients’ cohort not sufficient for a correct statistical analysis. Up-to-date we are performing an analysisi in the entire group of patients (more than 100) treated with metabolism guide irradiation that will permit a more accurate statistical analysis.
Round 2
Reviewer 1 Report
Comments and Suggestions for Authors
the comments are addressed
Author Response
Thank you very much for taking the time to review our manuscript. It has been a great opportunity to improve the quality of the paper and revise it completely. Thank you again for your contribution.
Reviewer 2 Report
Comments and Suggestions for Authors
I remain very ambivalent about reading the new version of the manuscript. I believe very strongly in the special effect of "lattice" radiotherapy, and in fact, for me it is one of the most interesting radiotherapy innovations in recent years. I greatly appreciate the further improvement of the paper and the work/discussion of the authors overall. However, I still miss the simple but well-founded objective/theoretical message to the expert readers.
As I mentioned in my previous report, it is not surprising that in case of high-dose SBRT (here: “biology-based neoadjuvant boost RT”) the surrounding tumorous/normal tissues receive relatively low doses (0,5-1-2-3Gy) of irradiation. SBRT is characterized by relatively sharp decrease in doses in volumes over PTV. This kind of dose gradient could be reached by delivering multiple/moving fields arrangement. The result is the relatively high volumes that receive relatively low doses. So, the result of this treatment planning/dose distribution study is obvious. Nevertheless, this low dose RT caused tissue damage could explain the special immunological and clinical response.
Overviewing the detailed RT treatment data of the work, and considering the first treatment part, the median Dmean of GTV was relatively high, 3.14 Gy (see line 232), supposing a kind of high, cytotoxic/palliative median dose to the whole GTV. Moreover, the meaning of e.g. V2: the volume that receives minimally 2 Gy, so a part of V2 receives 3 Gy, 4 Gy, 5 Gy etc. The authors try to prove that a significant part of the GTV receives “low dose” (between 0,5 and 2 Gy) irradiation, since experimental data suggests that this dose range is responsible for the triggering the immune response (see lines 350-351). So, I recommend a kind of a “derivation” of dose/volume data and the determination of the volumes that receives “per definition” LDRT (e.g. between 0,5 and 2 Gy or between 1 and 2 Gy, etc.). These results would indirectly prove that a relatively great volume (20-40%) of GTV that receives LDRT could be responsible for the immune effect enhancement.
Additional remarks:
Abstract: Please sign/explain the meaning of V0,5-V1-V2-V3.
Introduction: Lines 61-63: “assuming that the 18F-FDG uptake could be higher in capacity of hypoxic cancer cells with respect may be superior when compared with their normoxic one. counterparts” Maybe it is a perfect idea, but I miss some reference after this statement.
Discussion: Lines 310-329: The authors explain the possible cytotoxic effect of LDRT. I miss some references here.
Discussion: Lines 332-338: I suggest continuous description instead of enumeration, since these are general tumor-biological issues/considerations and are independent from the effect of LDRT.
Discussion: Line 470.: I suggest inserting “robust clinical data” into the sentence, since the original Lattice01 study provided important (and basic) clinical data
The number of vertices during SBRT: median 1 in the Abstract, median 2 (?) in the Results section (line 218), later in the Results section we can find the vertex data of 60 pts (lines 218-219), however the authors elaborated the treatment plans of 62 pts. Please clarify it.
Reviewer 3 Report
Comments and Suggestions for Authors
All concerns are addressed; no further comments.
Author Response

(The authors gave the same response as above.)
